# Revisiting Membrane Microdomains and Phase Separation: A Viral Perspective

**DOI:** 10.3390/v12070745

**Published:** 2020-07-10

**Authors:** Prabuddha Sengupta, Jennifer Lippincott-Schwartz

**Affiliations:** Howard Hughes Medical Institute, Janelia Research Campus, Ashburn, VA 20147, USA; lippincottschwartzj@janelia.hhmi.org

**Keywords:** HIV membrane, phase separation, protein sorting

## Abstract

Retroviruses selectively incorporate a specific subset of host cell proteins and lipids into their outer membrane when they bud out from the host plasma membrane. This specialized viral membrane composition is critical for both viral survivability and infectivity. Here, we review recent findings from live cell imaging of single virus assembly demonstrating that proteins and lipids sort into the HIV retroviral membrane by a mechanism of lipid-based phase partitioning. The findings showed that multimerizing HIV Gag at the assembly site creates a liquid-ordered lipid phase enriched in cholesterol and sphingolipids. Proteins with affinity for this specialized lipid environment partition into it, resulting in the selective incorporation of proteins into the nascent viral membrane. Building on this and other work in the field, we propose a model describing how HIV Gag induces phase separation of the viral assembly site through a mechanism involving transbilayer coupling of lipid acyl chains and membrane curvature changes. Similar phase-partitioning pathways in response to multimerizing structural proteins likely help sort proteins into the membranes of other budding structures within cells.

## 1. Introduction

Protein sorting is ubiquitous in biological membranes, involved in such diverse physiological phenomena as formation of transport carriers, biogenesis of extracellular vesicles, and receptor-mediated signaling [1,2,3]. The mechanistic basis underlying protein sorting has long been a topic of major research, with protein coats playing key roles [4,5,6,7,8]. Only recently, however, has the relevance of lipid-based phase partitioning to protein sorting been recognized [9,10,11,12]. This appreciation emerged from observations in model membrane systems showing selective protein sorting in response to a protein’s partitioning preference for specific lipid phases [13,14,15,16,17]. More recently, lipid-based phase partitioning in physiological systems has been demonstrated, for example, on the yeast vacuolar surface, where microscopically visible lipid-ordered and lipid-disordered membrane domains containing distinct sets of proteins are observed [18,19]. However, it has been challenging to detect such lipid-based phase separation and protein sorting in intact mammalian membranes under physiological conditions, possibly because the spatiotemporal scales of such events have been inaccessible to analysis by the available techniques [10,20,21]. In this review, we discuss how the investigation of protein sorting during human immunodeficiency virus (HIV) biogenesis has clarified how lipid-based protein sorting can operate in mammalian cell membranes [22].

Retroviruses, such as HIV, assemble at the plasma membrane (PM) of host cells and subsequently bud out as spherical viral particles. During this process, the assembly site membrane patch attached to oligomerized HIV structural protein, Gag, is transformed into the outer membrane of the released virus particle. This transformation is crucial for the virus to survive, infect other cells, and exhibit tropism [23,24]. Surprisingly, proteomic studies revealed that the expression level of certain proteins on the host cell PM is not correlated to the amount of proteins incorporated in the HIV outer membrane [23,25,26]. Likewise, the HIV outer membrane has a lipid composition distinct from the bulk lipid composition of the originating PM, being enriched in cholesterol, sphingolipids, and other lipids with long, saturated acyl chains [27,28,29]. These observations hinted at some type of lipid-based phenomenon operating to selectively incorporate proteins and lipids into the viral membrane during the assembly process [30,31,32,33,34]. This review expands on this idea, summarizing recent data that support a model of HIV membrane sorting involving lipid-based phase partitioning. We also discuss a model for HIV assembly site phase separation that involves the joint activities of transbilayer coupling of lipid acyl chains and membrane curvature changes.

## 2. Real-Time Analysis of HIV Assembly Reveals Time-Dependent Formation of Specialized Domains Driven by Gag Oligomerization and Phase Partitioning

The assembly of HIV involves the expansion of a protein lattice at the cytoplasmic face of the PM by progressive oligomerization of HIV Gag [35,36], a process spanning a time window of 10–15 min [22,37,38,39]. The analysis of the membrane from released HIV particles revealed it to be distinct in composition from the bulk PM. This suggested sorting of PM proteins and lipids into the HIV membrane during viral assembly. Two competing models have been proposed to explain such sorting [30,31,32,34]. The first model is passive incorporation by assembly of HIV particles at pre-existing PM domains with specialized protein and lipid composition. The second model is active remodeling of the local membrane by assembling HIV particles to create a domain with specific protein content, which eventually transforms into the HIV membrane. To distinguish between these two probable mechanism(s), recent work used quantitative live cell imaging and analysis tools to visualize changes in the HIV assembly site membrane across the entire time window of single virus assembly [22]. The real-time evaluation of HIV assembly site formation revealed that HIV Gag is not targeted to preexisting membrane domains with composition similar to the viral outer membrane. Rather, HIV Gag creates an ordered membrane domain at the assembly site and orchestrates protein sorting through a mechanism involving lipid-based phase partitioning (Figure 1).

In the above study [22], the protein and lipid composition of the HIV assembly site membrane was found to continually change throughout the assembly process, with three stages of HIV membrane assembly identified. The initiation of multimerization of membrane-anchored Gag molecules at the PM marked the first stage. This correlated with the recruitment of ordered lipid phase-preferring lipids (including sphingomyelin) and lipid-anchored proteins, such as glycosyl phosphatidylinositol (GPI)-anchored CD59, to the assembly site from the surrounding bulk PM. An ordered lipid phase-preferring transmembrane protein, murine leukemia virus envelope protein (MLV-Env), also redistributed to the assembly site from the bulk PM at this time. The second phase of the assembly process was characterized by specific disordered phase-preferring proteins becoming depleted from the assembly site, likely resulting in an even greater ordering of lipids at this site, with continued recruitment of ordered lipid phase-preferring proteins. The third, final stage of assembly was characterized by the completion of Gag oligomerization and a high membrane curvature at the assembly site. This led to the recruitment at the assembly site of other proteins from the surrounding bulk PM, including dual-membrane lipid-anchored proteins, such as tetherin [40,41,42].

This real-time evaluation of assembly site remodeling suggests an important distinction between Gag-induced phase separation at the HIV assembly site and lipid-based phase separation in model systems. Lipid-based phase separation processes in model membrane systems are characterized by switch-like, first-order phase transitions [14,43]. HIV assembly site phase separation, in contrast, proceeds in a series of stages extending over minutes (Figure 1) [22]. There is continuous recruitment of ordered phase-preferring proteins and lipids to HIV assembly site, with late-stage depletion of disordered phase-preferring proteins. This sequence of protein and lipid redistribution events suggests that there is increased ordering of assembly site lipid acyl chains with progress in viral assembly. Lipid acyl chain-ordering sensors, such as Laurdan and push–pull pyrene dye [44,45], offer useful tools for confirming the progressive ordering of the assembly site domain in future studies. The compositional changes during the prolonged assembly process amplify the differences in physical properties between the assembly site membrane and the bulk PM. This, in turn, helps in further sorting of additional proteins as HIV assembly progresses.

## 3. Sequential Protein Sorting May Help Explain Differences in Protein Concentrations in the Viral Membrane

A novel concept emerging from the real-time analysis of single virus assembly is the temporal structuring of the assembly process into distinct windows of protein sorting. The sequential sorting of different proteins between assembly site domain and bulk PM contributes to the continuously changing composition of the assembly site. These temporally distinct sorting events can be utilized for controlling the concentration of proteins in the HIV membrane by confining their recruitment to specific time windows. For example, early-recruited proteins might be expected to become more numerous in the virion membrane compared to late-recruited proteins. Consistent with this possibility, the envelope protein of the gammaretrovirus MLV (MLV-Env), recruited early, has a high number of proteins in the HIV membrane [22]. In contrast, only a small number (~ 14+/-7) of HIV envelope proteins (HIV-Env) is included in the HIV membrane [46,47]. This could be achieved by timing the recruitment of HIV-Env to coincide with the late stage of assembly [48], due either to a preference of Env for the unique morphology of the assembly site or to the delivery of Env to the PM in the late stage of assembly (Figure 2) [49]. We anticipate that a real-time evaluation of fluorescently labeled HIV-Env [50] distribution at single virus assembly sites will help to clarify if the time of recruitment to the assembly site is a determinant of the eventual density of Env within the HIV membrane.

The envelope proteins of lentiviruses such as HIV have a significantly longer cytoplasmic tail compared to gammaretroviruses like MLV [51]. As a negative Gaussian curvature at the neck is juxtaposed to a positive Gaussian curvature of the viral bud head, this geometrical landscape could impair the diffusion of proteins with long cytoplasmic tails, like HIV-Env, across the neck region, and thereby limit the incorporation of these proteins at a late stage of assembly. In such a scenario, the geometrical constraints of the neck region in conjunction with the lipid phase characteristics of the assembly site would help dictate the number of proteins being incorporated in the HIV membrane. The long cytoplasmic tail of lentivirus envelope proteins might also encounter significant steric interactions with the underlying Gag lattice at the viral assembly site, potentially making HIV-Env proteins at assembly sites highly immobile [52,53]. Another factor modulating HIV-Env incorporation could be the steric fitting of the cytoplasmic tail of HIV-Env within specific locations of the Gag lattice [54,55]. These steric interactions could limit the number of Env proteins in the viral membrane and give rise to specific distributions of Env proteins into the bud head versus neck regions [48,56].

## 4. Role of Transbilayer Coupling in Assembly Site Phase Separation

HIV Gag uses two specific structural elements to bind to PM: an N-terminus myristoyl lipid anchor and a stretch of basic amino acids within the highly basic region (HBR) of its matrix domain. Gag is anchored to the PM by the insertion of the myristoyl chain into the inner leaflet [57,58,59] and the interaction of the HBR with negatively charged lipids such as phosphatidylinositol 4,5-biphosphate (PIP2) and phosphatidylserine (PS) [60,61,62]. Thus, a direct physical interaction between Gag and the PM is confined to the PM inner leaflet via these two hydrophobic and electrostatic interactions. However, since the ordered lipid domain generated by Gag multimerization spans the two leaflets of the PM and involves remodeling of both leaflets, as evidenced by the enrichment of outer-leaflet anchored proteins and sphingolipids at the assembly site along with the reorganization of inner-leaflet anchored proteins [22], Gag–PM interactions must somehow mediate phase separation across both PM leaflets.

An appealing possibility for how Gag multimerization at the inner PM leaflet could impact events on the outer PM leaflet could be through Gag’s interactions with PIP2 and PS. High-affinity interactions between HIV Gag and PIP2 are expected to cluster PIP2 at the viral assembly site. This is supported by recent studies in model membranes and mammalian cells showing that PIP2 is concentrated at sites of Gag multimerization [63,64]. Acidic lipids like PS should similarly be immobilized at the HIV assembly site via their interactions with the HBR of Gag [29,65]. Crucially, however, both mammalian PIP2 and PS have a long, saturated acyl chain at the sn1-position and an unsaturated acyl chain at sn2-position [66,67,68]. We propose that the sn1-saturated acyl chains of immobilized acidic lipids at the viral assembly site undergo direct transbilayer interactions with the long, saturated acyl chains of outer leaflet lipids and GPI-anchored proteins (Figure 3). Such “transbilayer coupling” would draw in and concentrate outer leaflet lipid molecules into the assembly site. The immobilization of lipids across the two leaflets would compensate for the entropic penalty for local phase separation and facilitate the de-mixing of assembly site lipids from the surrounding bulk PM. The expansion of the viral assembly platform is mediated by the progressive multimerization of Gag. This would increase the immobilization/concentration of inner leaflet acidic lipids at the viral assembly site. The presence of multiple saturated acyl chains in close proximity within the assembly site domain will amplify the effects of transbilayer coupling and facilitate both enlargement and compositional differentiation of the assembly site lipid microdomain. 

The above transbilayer coupling model invokes the involvement of outer leaflet lipids and lipid-anchored proteins in initiating the transition of the assembly site into an increasingly ordered lipid domain, which is supported by the early recruitment of GPI-anchored proteins and other order-preferring lipids to the assembly site [22]. A different transbilayer coupling mechanism mediated by actin–PS interactions has recently been proposed in the generation of nanodomains of GPI-anchored proteins [68]. However, these PS–actin interactions were extremely small (~5 nm) and relatively short-lived (lasting for ~0.1–1.0 seconds), in contrast to the stability of the assembling virus platform, which lasts for many minutes and expands to cover a significantly larger area (the average diameter of immature HIV particle is ~133 nm [69]). 

## 5. Role of Membrane Curvature in Assembly Site Protein Sorting

As viral assembly progresses, the assembly site membrane juxtaposed to the expanding Gag platform increases its curvature [70,71]. Initial studies using Gag platforms with low curvature indicated that change in membrane curvature is necessary for sustaining progressive changes in the composition of the assembly site [22,72]. When the assembly site is unable to acquire high membrane curvature, the late-stage protein redistribution at the viral assembly site is blocked. We discuss two possible mechanisms for membrane curvature-mediated protein sorting during late stages of viral assembly.

One possible mechanism involves the curvature of the assembly site membrane facilitating protein sorting by imposing geometrical constraints to the partitioning of proteins and lipids. The two adjacent membrane areas, bulk PM and assembly site, present contrasting curvatures to the membrane proteins, which can sort between the two environments based on their molecular shapes [73,74]. Additionally, the local curvature imposed by the underlying Gag platform can also directly contribute to changes in the lipid composition of the assembly site membrane. The curvature that a section of the membrane can assume is dependent on the molecular shape, packing, and identities of the lipids in the opposing leaflets. The shape-based geometrical preference of lipids, when combined with favorable interactions between lipids, can drive the sorting of lipids between assembly site and bulk PM. Such lipid sorting can be especially strong at the viral assembly site, where the local membrane composition is likely to be close to that promoting phase separation [75,76,77]. The selective sorting of lipids will amplify the difference in physical properties between the assembly site and the bulk PM, and thereby enhance the sorting of proteins.

The curving of the assembly site domain can also contribute to protein sorting by a second mechanism that involves modulation of the energy at the interface of assembly site and bulk PM. The mismatch in physical properties between the assembly site and the bulk PM would give rise to an interfacial energy, called line tension, at the boundary between the two membrane regions [78,79]. The boundary energy, a measure of total interfacial energy penalty arising from line tension, is proportional to the length of the phase boundary between the two membrane environments (i.e., the perimeter of the assembly site domain). A key feature of the viral assembly process that could lead to increasing boundary energy during viral membrane assembly is the increasing difference in the membrane physical properties between the assembly site and the surrounding PM, which would increase the line tension at the assembly site boundary [78,80]. The energy penalty associated with increasing boundary energy can be significant for the nanoscale viral assembly domain. The curving of the assembly site domain away from the plane of the PM would decrease the interphase boundary length between the two membrane regions and help reduce the boundary energy [81,82,83]. The decreased boundary energy can stabilize the assembly site domains, and facilitate additional protein sorting and further changes in assembly site composition with progress in viral assembly. The curving of the assembly site domains can also facilitate their expansion without coalescence of neighboring domains [84].

## 6. Overall Scheme for Protein Sorting during HIV Assembly

The above results and ideas suggest a new way of thinking about protein sorting during retroviral budding that is based on active-phase partitioning of proteins between bulk PM and an assembly site membrane domain. In this scheme, the multimerization of an extrinsic membrane element, HIV Gag, at the inner leaflet of the bilayer initiates phase partitioning by ordering the inner leaflet lipids it contacts (Figure 3). Inner leaflet ordering subsequently causes outer leaflet ordering as a result of physical interactions between the long, saturated acyl chains of outer leaflet lipids and the acyl chains of immobilized inner leaflet lipids. The resulting transbilayer coupling of acyl chains then enables lipids and proteins on both leaflets of the bilayer to sort into the assembly site, based on their affinity for being in an ordered lipid environment. Additionally, the increasing curvature of the assembly site domain facilitates continuous protein sorting during the assembly process by stimulating the shape-based sorting of proteins and lipids between the bulk PM and the assembly site domain. The assembly site curvature can also facilitate protein sorting by modulating the interfacial boundary energy. Through these progressive compositional and membrane curvature changes, the assembly site differentiates and acquires the composition of the HIV membrane.

## 7. Conclusions

A major challenge in studying protein sorting in biological membranes is that the spatiotemporal scales and location of the sorting events often make them inaccessible for direct visualization. Two features of the HIV assembly process make it a convenient platform for clarifying the mechanisms underlying the sorting of membrane proteins. First is the relatively slow kinetics of the HIV assembly process (~10–15 min), and second is the clear spatial demarcation of the assembly site by the oligomerizing Gag lattice. Results from real-time analyses of HIV assembly, as discussed in this review, have helped identify lipid-based partitioning as a major driving force for protein sorting during HIV assembly. We propose that phase partitioning and protein sorting at the HIV assembly require both transbilayer coupling and curvature changes in the membrane.

We anticipate that specific features of the proposed transbilayer model are likely shared by protein sorting events at other locations within mammalian cells. Transbilayer coupling can be initiated if the following two criteria are satisfied: (i) immobilization and local ordering of lipids in one of the leaflets across a significant spatial scale and (ii) presence of lipids and lipid-anchored proteins in the opposite leaflet that can interdigitate with the immobilized lipids. Once transbilayer coupling is established, the transmission of lipid ordering across the two leaflets can facilitate local phase separation to drive protein sorting by a partitioning mechanism. The immobilization of lipids for transbilayer coupling can occur by asymmetric interaction of coat proteins with one of the leaflets of the bilayer [4,5,8,85], by close adhesion of adjacent bilayers at organelle contact sites [86,87], or by coupling of lipids to immobile cytoskeletal elements via protein bridges [68,88].

The contribution of membrane curvature to the HIV sorting process is also significant. The assembly site membrane acquires high curvature as the underlying Gag lattice curves to form a spherical particle. Consequently, the viral assembly process is coupled to membrane mechanics and provides a platform for studying how membrane shape collaborates with specific protein-lipid interactions to drive membrane compartmentalization. As described in this review, bilayer coupling working in conjunction with membrane curvature is likely to be essential for the final differentiation of the viral particle membrane. Membrane curvature can help to amplify the assembly site protein sorting by inducing shape-based sorting of molecules and by reducing the boundary energy. Additionally, high membrane curvature could create a kinetic trap for retaining proteins sorted into the viral membrane.

The two mechanisms for assembly site phase separation and protein sorting proposed in this review, viz. transbilayer coupling and membrane curvature, are certainly not exhaustive of the possible biophysical principles involved in membrane remodeling and protein sorting during viral assembly. Rather, we anticipate that the concepts emerging from our study and the proposed mechanisms will serve as a starting point for more in-depth investigation that will reveal how additional mechanisms collaborate to drive sorting during viral assembly. For example, electrostatic interactions, as indicated in a recent study [89], could play a significant role in how specific proteins redistribute into HIV assembly sites. Since interactions with negatively charged acidic lipids are often involved in the membrane anchoring of structural proteins of retroviruses and other enveloped viruses, the electrostatic potential at the viral assembly sites is likely to be a general mechanism for modulating the association of proteins with assembly site membranes. Another likely player in protein sorting into the viral assembly site could be tetraspanin-enriched microdomains (TEMs), which are known to partition into viral assembly sites [90,91,92]. The protein content of viral assembly sites is also likely to be modulated by direct interactions of host proteins with viral accessory proteins. For example, Vpu, an HIV accessory protein, has been reported to sequester tetherin from HIV assembly sites [93,94]. The real-time analysis of the redistribution of these proteins at single HIV assembly sites and the identification of their interaction patterns will reveal how they contribute to the final composition of the HIV membrane.

In conclusion, given the role of phase separation in protein sorting into the HIV membrane, it will be interesting to see if this process is involved in other sorting steps within cells, such as the coat-dependent sorting of proteins into intracellular transport carriers. Indeed, phase partitioning could well be working in conjunction with coat proteins for specifying the unique cargo delivered into vesicular transport intermediates [5,85,95].

## Figures and Tables

**Figure 1 viruses-12-00745-f001:**
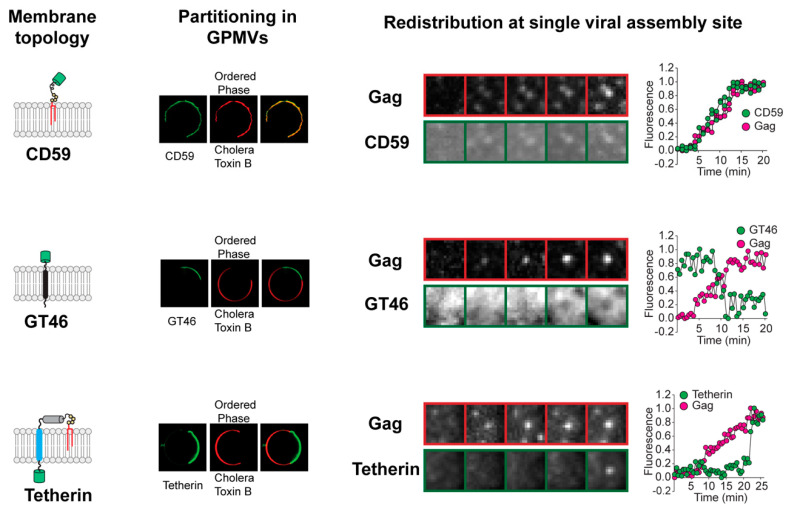
Redistribution of plasma membrane proteins at single viral assembly sites. The kinetics and direction of protein redistribution between the viral assembly site and the bulk plasma membrane are dictated by their affinity for ordered lipid membrane. The lipid phase preference of proteins with distinct membrane-anchoring elements (left column) is revealed by their partitioning between coexisting liquid phases in phase-separated giant plasma membrane vesicles (GPMVs) (middle left column). The Subunit B of Cholera Toxin marks the ordered phase in GPMVs. The time-series micrographs on the right show the redistribution of the proteins at single virus assembly sites. Ordered lipid phase-preferring proteins, such as glycosyl phosphatidylinositol (GPI)-anchored CD59, are continuously recruited to the assembly site during the entire length of the assembly process. The disordered phase-preferring transmembrane protein GT46, in contrast, is depleted from the assembly site during the middle phase of assembly. Tetherin, with dual membrane anchors, is recruited at the end of the assembly process when the viral assembly site has acquired a unique geometry with the viral bud attached to the plasma membrane via a highly curved neck region. Real-time protein redistribution data highlight novel features of the protein sorting process during viral assembly. The oligomerization of HIV Gag initiates the sorting of proteins into the assembly site membrane. This indicates that Gag actively creates the membrane domain that acts as a platform for protein sorting. Furthermore, the unique content of the viral membrane is achieved through a protracted process. The viral assembly site undergoes continuous remodeling, marked by sequential redistribution of proteins between the viral assembly site and the surrounding bulk plasma membrane. Scale bar, 5 µm. Images adapted from reference [20].

**Figure 2 viruses-12-00745-f002:**
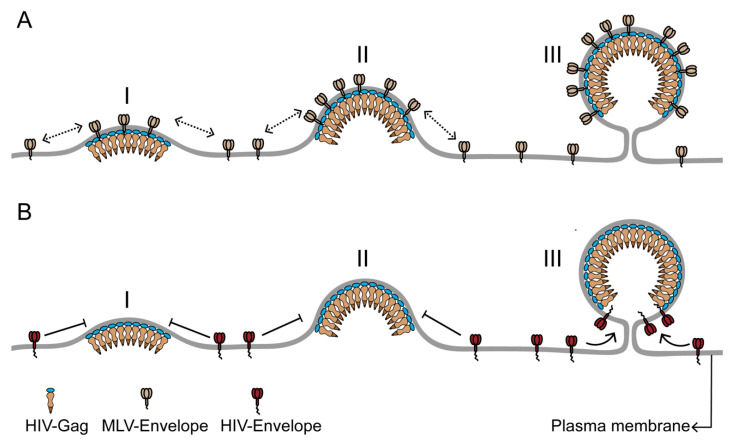
The differential incorporation of envelope proteins into the HIV membrane can result from differences in the time of recruitment to the HIV assembly site. (**A**) Murine leukemia virus envelope protein (MLV-Env) has strong affinity for the HIV assembly site due to its preference for ordered membrane phase and its small cytoplasmic domain. Consequently, MLV-Env is continuously recruited to the assembly site during the entire assembly process (stages I–III) and is enriched in the HIV membrane. (**B**) HIV envelope protein (HIV-Env), with a long cytoplasmic domain, is likely to be sterically excluded from the assembly site during the early and middle stages (stages I and II) of assembly. A low density of HIV-Env on the plasma membrane could also contribute to its absence from the assembly site during the assembly process. HIV-Env is recruited to the assembly site at the terminal stage of viral assembly (stage III), when the highly curved assembly site acquires a unique morphology, with a spherical bud attached to the PM by a narrow neck. The HIV Gag lattice at the end of the assembly process is arranged as a continuous but incomplete sphere in the virus head, with no Gag bound to the section of assembly site membrane adjacent to the neck of the viral bud. This Gag-free section of the assembly site membrane can accommodate the long cytoplasmic domain of HIV-Env without steric interactions. The limited Gag-free membrane space juxtaposed to the neck region could also help to limit the number of HIV-Env proteins in the viral membrane. The Figure is not drawn to scale.

**Figure 3 viruses-12-00745-f003:**
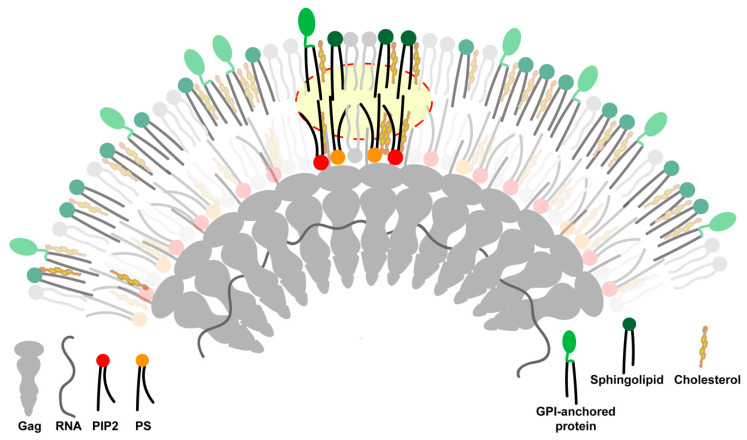
Transbilayer coupling model for viral assembly site phase separation. The oligomerizing Gag platform clusters and immobilizes inner leaflet acidic lipids, phosphatidylserine (PIP2), and phosphatidylserine (PS) at the viral assembly site. The ordering of the inner leaflet is transmitted to the outer leaflet via transbilayer interactions. The long, saturated acyl chains at the sn1-position of the acidic lipids can physically interact with the long, saturated acyl chains of outer leaflet lipids and lipid-anchored proteins (highlighted by the yellow backdrop). The resulting transbilayer coupling leads to the recruitment of outer leaflet sphingolipids and glycosyl phosphatidylinositol (GPI)-anchored proteins to the viral assembly site. The clustering of lipids and lipid-anchored proteins in both leaflets of the assembly site decreases the entropy of the mixing of assembly site lipids. This eventually triggers the phase separation of the viral assembly site from the surrounding bulk plasma membrane. The Figure is not drawn to scale.

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
