# Peer review of "Revisiting Membrane Microdomains and Phase Separation: A Viral Perspective"

_viruses, 2020, doi:10.3390/v12070745_

Round 1

Reviewer 1 Report

See attached file for comments and suggestions

Author Response

We thank the reviewer for raising some important points. We have made revisions in the manuscript to address the issues, which we think has substantially improved our manuscript. Below, we provide a comprehensive response to the issues raised by the reviewer.

In paragraph 2.2, a part of the model is presented that does not seem to be univocally supported by experimental evidence, and where alternative interpretations of the data are not given. The authors start from the observation that the signal both from cholesterol and SM increases in intensity over time, a trend observed also for liquid-ordered preferring proteins and for HIV Gag. The conclusion reached is that lipid and proteins are continuously enriched at the domain site, which seems to imply that the domain size remains roughly the same over the whole process, and that lipids and proteins are continuously partitioning more and more in this stable domain. This view is further confirmed by the authors in line 135, where they state “Unlike the switch-like first order phase transitions observed in model membrane systems, the assembly site phase separation is a protracted event spanning the entire assembly event.”

However, an alternative interpretation is, for example, that the domain size increases resulting in higher fluorescence signal from Gag and labelled proteins. Moreover, there is no theoretical prediction or experimental evidence cited to support the notion of slow-forming domains (not growing kinetics, but progressive ordering over long times) as suggested for viral assembly sites?

---- The reviewer is right that the progressive increase in signal of order-preferring proteins/lipids  during the assembly process can be due to enlargement of the sub-diffraction Gag platform, with no net increase in density of these order-preferring proteins and lipids. However,  we would like to point out that we also observe depletion of disorder-preferring proteins only at later stages of assembly (after at least 50% of the assembly has occurred), and recruitment of tetherin only towards the end of assembly. These results demonstrate that the assembly site undergoes compositional changes during the length of the assembly process, and the protein sorting/phase separation process is indeed a protracted event.

     Furthermore, the depletion of disorder lipid phase-preferring proteins at late stages of assembly is consistent with the assembly site acquiring increased ordering at late stage  when the disordered preferring proteins can no longer be accommodated. Taken together, we think that it is a reasonable hypothesis that the assembly site gets more ordered over time. We agree with the reviewer that imaging the assembly site domain in real-time with lipid order sensors, such as Laurdan, is required to confirm that the assembly site domain ordering is actually increasing with progress in assembly. We have revised the manuscript to mention more explicitly that the idea that the assembly site is progressively ordered  needs to be confirmed by future experiments (please see Section 2, Paragraph 3).

We would like to note that imaging of single virus assembly in real-time with lipid-order sensor dyes is challenging.  The single virus assembly can only be visualized in real-time by time-series imaging under  strict TIRF conditions. It is challenging to image most of the lipid-order sensing dyes under these imaging conditions owing to their low photostability when imaged with short wavelengths. The lipid-order sensor dyes work best with 2-photon excitation or for life-time imaging, conditions where it is challenging (if not impossible) to image single viral assembly in real-time.  However, we have made some progress in imaging newly developed lipid-order sensing dyes (with longer emission wavelengths) and evaluating life-time of the dyes at single viral assembly sites. We anticipate that we will soon have data regarding the ordering of lipid acyl chains during viral assembly (the Covid-19 pandemic has put a temporary hold to our experiments).

The authors also state that “Lipid acyl chain ordering sensors confirmed that the recruitment of these order preferring molecules is accompanied by increased ordering of lipid acyl chains at the assembly site membrane” (lines 89 – 91), however in the original paper (ref 13 of the manuscript) there is no experimental data using classical lipid packing sensors such as Laurdan or Di-4. Instead, the authors employed fluorescently labelled cholesterol or sphingomyelin, which, although highly enriched in liquid ordered domains, do not provide in itself a measure of the lipid order. In general, the assumption that the assembly site becomes more and more ordered is not supported by direct experimental evidence. The assumption that presence of cholesterol and SM is locally enriched over time can be disputed by the same argument of protein partitioning, i.e. from the fluorescence intensity is not possible to say whether there is indeed progressive enrichment under the same size, or increase in size below the diffraction limit.

---- Please see response above. 

In paragraph 3.1, the authors state that “Both mammalian PIP2 and PS have a long, saturated acyl chain at 1’-position. We propose that the long, acyl chains of immobilized acidic lipids at the viral assembly site would undergo direct transbilayer interactions with long, saturated acyl chains of outer leaflet lipids and GPI-anchored proteins”. It should be pointed out that the majority of PS lipids in the inner leaflet of plasma membranes have only one saturated acyl chain, whereas the second acyl chain is highly unsaturated (this is not mentioned in the main text, although it is hinted in the schematics of Fig 2, where we can see both PS and PIP having on long elongated tail and one “kinked” tail, typical of lipid with different degree of unsaturation between the two acyl chains), as also shown in a recent lipidomics analysis of biological membranes (see Lorent et al. Nature Chemical Biology 2020).

---- The reviewer is right in pointing out that mammalian PS and PIP2 have one saturated (18:0) acyl chain at sn1-position and an unsaturated acyl chain at sn2-position. The difference in saturation of the two acyl chains are represented in our model in Figure 2 with a straight and a bent acyl chain. In the revised manuscript, we have  mentioned in the text the presence of the unsaturated chain at sn2-position of PIP2/PS  (Section 3, paragraph 2).

It is not clear whether a single saturated acyl chain would be sufficient for strong transbilayer coupling, considering that also in reference 46 of the manuscript the majority of data used PS species with both saturated chains (18:0-18:0) and fewer measurements were provided for PS with mismatched acyl chains. The authors should perhaps discuss this issue and potentially add additional references to more completely cover the topic, or otherwise provide some insight or reasoning as to why this phenomenon would happen

---- As we discussed, Gag’s interaction with the PM is confined to the inner leaflet via (1) interaction between the PIP2/PS and the highly basic region of the matrix domain of Gag and (2) the insertion of C14 myristoyl chain of Gag into inner leaflet.  However, we observe robust recruitment of outer leaflet anchored GPI-anchored proteins and SM to the viral assembly site following initiation of Gag oligomerization. This indicates that the effects of inner-leaflet interaction of Gag is transmitted to the outer leaflet of the PM. The C-18 saturated chains of PIP2 and PS are likely candidates for coupling of the two leaflets. The oligomerization of Gag will cluster and immobilize  PIP2 and PS at the assembly site, creating a local domain enriched in PIP2 and PS.  Under such conditions, we think that  multiple 18:0 acyl chains present in close proximity within the assembly domain will be able to induce transbilayer coupling. The immobilization of the PIP2/PS molecules by interaction with the underlying Gag lattice would provide a stable anchor for transbilayer coupling. As we  noted in the manuscript, the immobilization and concentration of PIP2/PS at the viral assembly site happens over a larger spatial scale and for a significantly longer time compared to the creation of nanodomains of GPI-anchored proteins by transbilayer coupling with PS. Consequently, we expect that the transbilayer coupling at the viral assembly site would be stronger and more stable.

We had discussed these features of the viral assembly site in the manuscript, but we think the rational was not completely clear, as pointed out by the reviewer. In the revised manuscript, we have expanded our discussion to explain the rationale behind our  hypothesis that PIP2/PS-mediated  transbilayer coupling contributes to viral assembly site phase separation (please see Section 4, paragraphs 2 and 3).

It might also be more informative if Fig 2 was drawn to scale. The Gag layer of immature HIV particles is at least twice the bilayer thickness.

---- The reviewer has rightly pointed out that the model has not been drawn to scale (the depiction of the bilayer and the Gag). The Gag layer is approximately 5 times thicker than the lipid bilayer, with the Gag layer being ~22-25 nm in thickness while the lipid bilayer is ~4-5 nm in thickness ( J Virol. 2001 Jan; 75(2): 759–771). In our model, it is not necessary to depict the Gag and the bilayer to scale (i.e. have the Gag as 5 times thicker than the bilayer) since we are only interested in the interaction surface between Gag and the membrane bilayer; drawing the Gag layer as 5-times thicker than the bilayer will only take up space in the figure with no useful information. We have included a note in the figure legend of the  revised manuscript to mention that the Gag and bilayer is not drawn to scale.

Assuming that indeed there is compositional evolution over time in the domain, the statement that increasing line tension would arrest such compositional change, and halt further phase separation, deviates strongly from accepted models for domains growth kinetics and the role of line tension in regulating phase separation in general. It is well accepted, both theoretically and experimentally (see for example Garcia-Saez et al., JBC, 2007 and Usery at al., Biophysical Journal,2017), that positive line tension actually is a driving force for domain expansion and phase separation between lipids, and that rafts or domain cannot exist for negative line tension values.

Also at line 229 “By countering the increase in line tension, the increasing curvature would allow further changes in assembly site membrane physical properties. This would amplify the differences between assembly site membrane and bulk PM, and, trigger increased protein sorting with progress in assembly.”

---- We would like to note that our model does not contradict  accepted models for role of line-tension in expansion of membrane domains and phase separation. We are sorry that our description was not clear and has given the impression that we are postulating that lowering line-tension drives the phase separation of assembly site. As we discuss below, we are proposing that the lowering of total boundary energy (not line-tension) due to curving of the assembly site domain can help in increased sorting of proteins. We have revised Section 5 (discussion about curvature mediated protein sorting) to provide a clearer description about the possible curvature-mediated  mechanisms.

Our results clearly show that the increased curvature of the assembly site is required for middle- and late-stage proteins sorting events. However, how membrane curvature mediates protein sorting at assembly site is not yet clear, we are currently working to identify underlying mechanism/s. In this review, we have discussed possible mechanisms,  one of them being the geometrical constrains imposed by curvature and molecular-shape based sorting of protein and lipids (second paragraph of Section 5, revised manuscript).  In the third paragraph, we discuss how shortening of the interfacial boundary might help in increased protein sorting.

As the reviewer pointed out, line-tension can promote expansion of membrane domains by decreasing the total line energy of multiple nanodomains. The total boundary energy due to line-tension is proportional to the interfacial boundary length. Smaller nanodomains have higher boundary to area ratio compared to larger domains (as domain grow in size, the surface area increases to larger extent than the boundary, the surface increase is proportional to r^2 whereas boundary increase is proportional to r). Consequently, multiple nanodomains will have larger total boundary energy compared to fewer large domains with same  total surface area. For larger values of line-tension, the decrease in boundary energy dominates and larger domains are likely to be  preferred over multiple nanodomains.

However, the situation is different during viral assembly as the viral assembly site  is not  driven by spontaneous merger of membrane nanodomains to larger domains. The growth of the viral assembly domain is driven by  the progressive oligomerization of Gag, which leads to expansion of the underlying protein platform. This is supported by our data showing that all membrane remodeling and protein/lipid sorting events happen following initiation of Gag oligomerization. The growing size of the Gag platform translates into a larger domain containing greater number of immobilized PIP2/PS. In our model, the assembly site phase separation is initiated by transbilayer coupling between immobilized PIP2/PS and outer leaflet GPI-anchored proteins and lipids with long, saturated acyl chains. Thus, the size of the phase separated assembly site domain is dictated by the size of the underlying Gag lattice. 

The formation of the assembly site domain will also give rise to line-tension at the boundary of the domain due to difference in physical properties of the domain and the surrounding bulk PM. The line-tension is defined as the energy penalty per unit length of the domain boundary. The total boundary energy (total energy penalty due to line-tension) of a domain is proportional to the length of the  domain boundary, i.e. the perimeter of the domain. This boundary energy can be significant for nanodomains such as the assembly site domain in the initial stages of viral assembly. Now if the domain curves away from the membrane surface , as is the case for the viral assembly site, the length of the interfacial boundary will decrease. This will lead to decrease in total boundary energy even though the line-tension stays the same. We are proposing that this could allow the domain to acquire further compositional changes. This is because the increase in line-tension arising from additional compositional change of the domain can be compensated by the shortening of the domain boundary, thereby maintaining similar total boundary energy. We think this could be a contributing factor to the coupling of enhanced protein sorting with curvature.

We realize that our discussion in the original manuscript was not clear and apologize for any misunderstanding.

Lowering the line tension has been demonstrated by several experiments to inhibit domain size growth and stabilize domain size, so curvature should not in principle drive further phase separation. Interestingly, curving of domain (or buckling) has been shown to play a role not only in lowering the interfacial line-tension, but also by providing an energy barrier between curved domains to prevent their coalescence (Ursell et al., PNAS, 2009). In light of accepted theoretical models for line tension role in domain growth and curvature in domain coalescence, an alternative interpretation of the results would be that reducing the line tension via curvature prevents the assembly site to further increase their size and stabilizes the domain, and moreover allows for multiple neighboring assembly sites without incurring in the risk of them collapsing together (indeed, as shown in Fig 6 of reference 13 of the manuscript, two assembly sites can be seen in close proximity to each other)

---- Lowering the total boundary energy penalty (arising from line-tension) by increased curvature of a domain  can indeed inhibit domain growth. However, as we discussed above, we are not postulating that lowering line-tension promotes assembly site domain growth.

Our data shows that late stage protein sorting (depletion of disorder preferring proteins) happen only after assembly site has acquired significant curvature, indicating that increased curvature is required for late stage membrane remodeling. This supports our idea that curvature mediates further compositional change of assembly site at later stages of assembly.

We have primarily focused on assembly site composition changes in this review. We agree with the reviewer that curvature can have additional effects on the assembly site domain: increased curvature can help in delimiting the size of the assembly site and prevent coalescence of neighboring assembly sites. We would, though, like to point out  that the viral assembly site continues to grow even after it has started to curve: electron micrograph images show that  the increase in size of the assembly site happens concomitantly with increasing curvature of the assembly site. Thus, lowering boundary energy by curving the assembly site does not stop growth of assembly site domain immediately following increase in curvature of assembly site (even though new patterns of protein redistribution emerge as the curvature increases). However, at terminal stage of viral assembly, when the assembly site has maximum curvature and has acquired a spherical shape, the geometry of the assembly site can indeed help in stopping the growth of the assembly site.

Reviewer 2 Report

The review article "Revisiting lipid rafts and membrane microdomains: the viral point-of-view", by Prabuddha Sengupta and Jennifer Lippincott-Schwartz proposes a model, based upon convincing observations, where the building of the HIV protein enveloppe by aggregation of Gag proteins leads to the organization of a raft-like domain in the cell membrane close to the growing viral capsid. During the 10-15 minutes the capsid formation takes, a series of molecular recruitment events eventually leads to a virus outer membrane composition different from the average host cell plasma membrane composition, as observed.

The review is convincing and well written, and will probably deserve to be published. However, I have one major comment about the line tension argument developed in the paragraph starting on line 224. Indeed, I think that the authors make a confusion between the line tension and the line energy there. The line tension, expressed in Newtons, does not depend on the boundary length between the two distinct phases that the boundary separates. It is an intensive physical quantity that only depends on the nature of the two adjacent phases, see for example T.S.Ursell, et al., PNAS 106, 13301 (2009) in a similar context, or Julicher, Lipowsky Physical Review E 53, 2670 (1996) for an older work. It does not depend on the boundary length.

In the preceding paragraph, beginning one line 215, I am not completely convinced by the proposition that increasing line tension would lead to an  energy penalty sufficient to "deter enhanced phase separation". Indeed, the energy gain ensuing from phase separation is proportional to the domain surface, whereas the penalty is proportional to the domain perimeter, itself proportional to the domain surface square root. Of course, for a nanodomain, the boundary can become predominant because it is very small, but this should be quantified, at least grossly.

I also have some minor comments:
1. On line 170, I would replace "decrease the entropic penalty" by "compensate the entropic penalty". Indeed, there is an entropic cost associated with domain formation, compensated by a greater energetic gain.
2. In the last paragraph of section 3, some concrete examples of proteins or lipids going to the curved region would be welcome, in the HIV context.
3. I am not sure whether section 4 is very useful, it rephrases what was already stated above.

Once this is clarified, the paper can for sure be published in Viruses.

Author Response

We would like the reviewer for their constructive criticism and helpful suggestions. We have addressed the issues raised by the reviewer and made changes in the manuscript. We think this has helped to improve our manuscript.

The review is convincing and well written, and will probably deserve to be published. However, I have one major comment about the line tension argument developed in the paragraph starting on line 224. Indeed, I think that the authors make a confusion between the line tension and the line energy there. The line tension, expressed in Newtons, does not depend on the boundary length between the two distinct phases that the boundary separates. It is an intensive physical quantity that only depends on the nature of the two adjacent phases, see for example T.S.Ursell, et al., PNAS 106, 13301 (2009) in a similar context, or Julicher, Lipowsky Physical Review E 53, 2670 (1996) for an older work. It does not depend on the boundary length.

As the reviewer has pointed out, we were not precise in how we used the term line-tension. The terms line-tension and line-energy had been used interchangeably in the original manuscript, we  thank the reviewer for pointing out the discrepancy and have corrected it in the revised manuscript. We have used the term boundary energy to represent the total energy at the edge of the domain and clearly distinguished the total energy from line-tension (energy penalty per unit length). Please see Section 5, Paragraph 3, of revised manuscript.

In the preceding paragraph, beginning one line 215, I am not completely convinced by the proposition that increasing line tension would lead to an  energy penalty sufficient to "deter enhanced phase separation". Indeed, the energy gain ensuing from phase separation is proportional to the domain surface, whereas the penalty is proportional to the domain perimeter, itself proportional to the domain surface square root. Of course, for a nanodomain, the boundary can become predominant because it is very small, but this should be quantified, at least grossly.

We find that the curvature of the assembly site is necessary for increased protein sorting at later stages of assembly, though the mechanism/s for such sorting is not yet entirely clear. A likely mechanism for the late stage protein sorting is geometry/shape based sorting of molecules between the assembly site and the bulk PM based on their distinct curvatures.  This mechanism can be tested experimentally and we are in the process of designing and performing experiments to evaluate its possible involvement in assembly site protein sorting.

However, the Gag assembly sites are nanodomains (tens of nanometer in size) where the boundary energy, as the reviewer says, can play a significant role.  We are proposing that the curving of the assembly site domain might help in additional protein/lipid sorting (which would likely increase line-tension by amplifying the difference between assembly site and bulk PM ) by decreasing the overall boundary energy. The underlying oligomerized Gag lattice presumably provides the energy for curving the nano-Gag assembly domain. In such a scenario, we are assuming a shorter boundary perimeter (arising from the domain curving out of the plane of PM) can compensate for the increase in line-tension from additional compositional changes, and contribute to protein/lipid sorting. We are interested in learning if the reviewer thinks this a reasonable notion. We anticipate that future experiments will reveal the interplay of these factors in late stage protein sorting. We have revised the section on curvature to provide a more balanced (discussing the  geometry based mechanism in the first paragraph) and clearer discussion about these possible mechanisms. Please see Section 5 of revised manuscript.

I also have some minor comments:
      1. On line 170, I would replace "decrease the entropic penalty" by "compensate the entropic penalty". Indeed, there is an entropic cost associated with domain formation, compensated by a greater energetic gain.

We thank the reviewer for the suggestion and have made the change in the revised manuscript (section 4, paragraph 2).

  1. In the last paragraph of section 3, some concrete examples of proteins or lipids going to the curved region would be welcome, in the HIV context.

Proteins such as CD59 and MLV-Env are continuously recruited throughout the assembly process, so they are also recruited to assembly site with increased curvature. We assume that the reviewer is referring to proteins/lipids which are recruited only after the assembly site has acquired high curvature, but not during earlier stage of assembly. Till now, we have observed the recruitment of tetherin to late stage assembly site and depletion of disorder-preferring proteins (GG and GT46). We are now testing redistribution  other proteins and lipids and anticipate that we will be able to identify additional proteins and lipids that are recruited to late stage curved assembly site. We are also investigating if/how shape of proteins affect their partitioning between assembly site and bulk PM, we anticipate that we will soon have some clear evidence for shape-based sorting.

  1. I am not sure whether section 4 is very useful, it rephrases what was already stated above.

We thank the reviewer for the helpful suggestion. We have made substantial changes to this section (as well as preceding sections) in the revised manuscript and we think this has helped to eliminate some of the redundancies that the reviewer pointed out.

Reviewer 3 Report

The authors have written an outstanding review article on protein sorting during HIV biogenesis.

Recommendations are below :

  • Line 29-31: This sentence is hard to understand the way it is set up. Needs rewording/reshuffling of sentence structure
  • Line 36-37: Why has it been challenging? Is the technique not working or are the results not what are expected?
  • Line 79-80: Sticking to the term “viral membrane” might be better than intermixing it with “HIV membrane” for simplicity/ease of understanding
  • Line 273: Typo in section title
  • Line 290: Reference for functional activity of labelled Env would be helpful
  • Line 357: There is evidence of viral accessory proteins changing the landscape of viral assembly sites (Vpu displacing tetherin, see McNatt et al 2013, Pujol et al 2016). Mentioning how there are viral proteins other than Gag that help create the optimal budding environment at the PM is important to mention, especially if they are/will be excluded from your analyses
  • In the conclusion, more could be said about the future studies other than the goal of these studies. What techniques will be used?
  • It appears that there are lots of double spaces mixed in throughout.

Need to define abbreviations at first use (ex PM on line 44, PIP2 and PS on line 147, GPI, GPI-AP line 173).

Author Response

Comments and Suggestions for Authors

The authors have written an outstanding review article on protein sorting during HIV biogenesis.

We thank the reviewer for their positive response and for the helpful suggestions. We have addressed the issues raised by the reviewers.

Recommendations are below :

  • Line 29-31: This sentence is hard to understand the way it is set up. Needs rewording/reshuffling of sentence structure

We have revised the introduction section of the manuscript and we hope that he introduction section is now clearer and easier to follow.

  • Line 36-37: Why has it been challenging? Is the technique not working or are the results not what are expected?

The reviewer has raised a critical question. Lipid based sorting in biological membranes has been challenging to detect in live cells  possibly because the spatiotemporal scales of compartmentalization were inaccessible by the available microscopic techniques. However, the inability to directly visualize such lipid-based sorting events  also raised the possibility that they might not operate in live biological membranes in the same way as has been observed in model membrane systems. We have revised the Introduction section and have included additional  references.

  • Line 79-80: Sticking to the term “viral membrane” might be better than intermixing it with “HIV membrane” for simplicity/ease of understanding

We thank the reviewer for the suggestion and agree that we should try to stick to one term throughout the manuscript. Since we needed to specify that we are referring to HIV membrane in certain cases, we have replaced the term “viral membrane” and have used the term “HIV membrane”  in the revised manuscript.

  • Line 273: Typo in section title

We thank the reviewer for pointing out they typo. We have revised the section titles in the resubmitted manuscript.

  • Line 290: Reference for functional activity of labelled Env would be helpful

We have included references in the revised manuscript describing labeling of HIV-Env with fluorescent proteins inserted  within the V4/V5 loops, the labeled Env proteins appear to maintain their biological function (evaluated by their fusogenic activities) (Section 3, paragraph 1 of revised manuscript). We anticipate that this strategy of labeling will help in visualizing HIV-Env redistribution in real time and help clarify the mechanisms underlying their incorporation into HIV membrane.

  • Line 357: There is evidence of viral accessory proteins changing the landscape of viral assembly sites (Vpu displacing tetherin, see McNatt et al 2013, Pujol et al 2016). Mentioning how there are viral proteins other than Gag that help create the optimal budding environment at the PM is important to mention, especially if they are/will be excluded from your analyses

We thank the reviewer for pointing out an important aspect of viral assembly site remodeling. In the revised manuscript, we discuss the role of viral accessory proteins in modulating assembly site environment and have also included references. Please see Section 6, paragraph 4 of revised manuscript.

  • In the conclusion, more could be said about the future studies other than the goal of these studies. What techniques will be used?

We anticipate that real-time analysis of redistribution of various proteins, membrane lipid ordering and membrane trafficking at viral assembly sites with high-resolution imaging techniques will help in obtaining further insights about how HIV (and other enveloped viruses) attain their distinct protein and lipid composition. We have revised the manuscript extensively and have incorporated discussion about this strategy. Additionally, evaluation of mutant versions of Gag with different membrane anchoring elements and geometrical properties, and, structural studies of Gag-membrane interactions would be required for a comprehensive understanding of protein/lipid sorting during viral assembly.

  • It appears that there are lots of double spaces mixed in throughout.

We apologize for the discrepancies in spacing and thank the reviewers for pointing it out. We realized that the formatting of the manuscript got changed when we inserted the manuscript into the journal template. We have tried to maintain constant spacing in the revised manuscript.

Need to define abbreviations at first use (ex PM on line 44, PIP2 and PS on line 147, GPI, GPI-AP line 173).

We thank the reviewer for the helpful suggestion. We have defined the abbreviations in the revised manuscript.

Reviewer 4 Report

The review by Sengupta and Lippincott-Schwartz revisits lipid rafts and membrane microdomains in the context of HIV plasma assembly and protein sorting.  The focus is mainly how lipid-based protein sorting operates in biological membranes.  The functional importance and unique composition of HIV membrane is established.  However, the underlying mechanisms of protein and lipid sorting into viral membranes remained unclear.  The current review provides insight how HIV Gag creates a phase separated ordered with selective incorporation of proteins into the nascent viral membrane by lipid-based partitioning of proteins.  The proposed model will be a base for future studies to establish a clear understanding lipid-based partitioning of proteins in virus particle assembly, budding and maturation.  This is a well written review and informative.  It would help adding a Figure in section 5 with cartoon representation depicting the possible relationship of Env inclusion into the assembly membrane site along the proposed model.

Author Response

The review by Sengupta and Lippincott-Schwartz revisits lipid rafts and membrane microdomains in the context of HIV plasma assembly and protein sorting.  The focus is mainly how lipid-based protein sorting operates in biological membranes.  The functional importance and unique composition of HIV membrane is established.  However, the underlying mechanisms of protein and lipid sorting into viral membranes remained unclear.  The current review provides insight how HIV Gag creates a phase separated ordered with selective incorporation of proteins into the nascent viral membrane by lipid-based partitioning of proteins.  The proposed model will be a base for future studies to establish a clear understanding lipid-based partitioning of proteins in virus particle assembly, budding and maturation.  This is a well written review and informative. 

We thank the reviewer for the positive comments.

 It would help adding a Figure in section 5 with cartoon representation depicting the possible relationship of Env inclusion into the assembly membrane site along the proposed model.

We thank the reviewer for the helpful suggestion. We have not yet investigated the inclusion of the HIV-Env proteins. Some of the current ideas about mechanism/s of HIV-Env incorporation into HIV membrane are not inconsistent with our proposed model. However, an important feature of our model is the temporal structuring of the protein sorting process, with the timing of protein recruitment to viral assembly site potentially modulating amount of protein incorporated in the viral membrane. The role of curvature in protein sorting indicated by our data and model also introduces another potential determinant for protein incorporation.  We have included a cartoon representation of how these features of our model can modulate the incorporation of HIV-Env into HIV-membrane.
